# Exploring NLP pipelines for textual regression of prices

## Abstract

Natural language (NL) price regression is the task of predicting prices from text. Like other NL applications, NL price regression uses a pipeline that typically comprises four steps: preprocessing, tokenization, featurization and modeling. Each step has multiple options, from traditional and modern approaches, giving many possible pipelines. However, there is no work systematically comparing different combinations of these steps for NL regression. We systematically generate and evaluate hundreds of random valid pipeline configurations, including combinations not commonly studied. For example, approaches with Transformers for featurization and gradient boosted trees for modeling. Then, we evaluate these pipelines on two real datasets. These experiments reveal several interesting aspects of pipeline construction: i) BERT contextual featurization outperforms GloVE non-contextual featurization, ii) BERT featurization needs to be finetuned to outperform bag of words, with implications for resource constrained applications, iii) the variance associated with choosing steps upstream from modelling is comparable to that of selecting the model, and iv) vector embeddings (BERT and GloVE) perform worse than bag of words for GBDT models. This study provides systematic evidence highlighting the need for holistic pipeline optimization for price regression.

## 1 Introduction

Regression, the task of predicting numeric values from data, is an extensively studied machine learning problem. Regression has applications in finance (Ariyo et al., 2014), medicine (Boateng & Abaye, 2019), engineering Rhinehart (2016), energy (Jónsson et al., 2010) and many other areas. Regression algorithms typically require the input to be numeric. However, many applications are emerging that would benefit from numeric predictions from text. For example: in commerce to predict product prices from descriptions; in finance for stock price regression from reports; or in insurance for claim damage prediction from accident reports. This gives rise to the question of how to use natural language in price regression.

This work investigates construction of NLP regression *pipelines*. We use 'pipelines' to refer to *every* step of NL regression: preprocessing that modifies raw text; tokenization that maps text to tokens; featurization that maps tokens to numeric representation; and modelling that predicts numeric outputs given the numeric representation.

Based on systematic study, our contributions, to the authors' knowledge, are the first quantification of the relative importance of certain pipeline choices. Specifically, we empirically demonstrate that contextual vector embeddings outperform non-contextual vector embeddings for NLP regression on the datasets studied. We also provide quantitative evidence that: i) finetuning is required for BERT featurizers to achieve better accuracy than bag of words (BoW) featurizers, highlighting the implications for resource constrained applications, ii) that choices other than the model are as important as the model itself (i.e. the choice of preprocessing, tokenizer and featurizer is associated with variance in accuracy comparable to that of choosing the model), and iii) that gradient boosted decision tree (GBDT) models perform worse using vector embeddings than with bag of words featurization.

Section 2 presents background on the NLP pipeline steps that we study. Section 3 presents related work. Section 4 outlines the experimental procedure used in this study. Section 5 presents results and discussion.

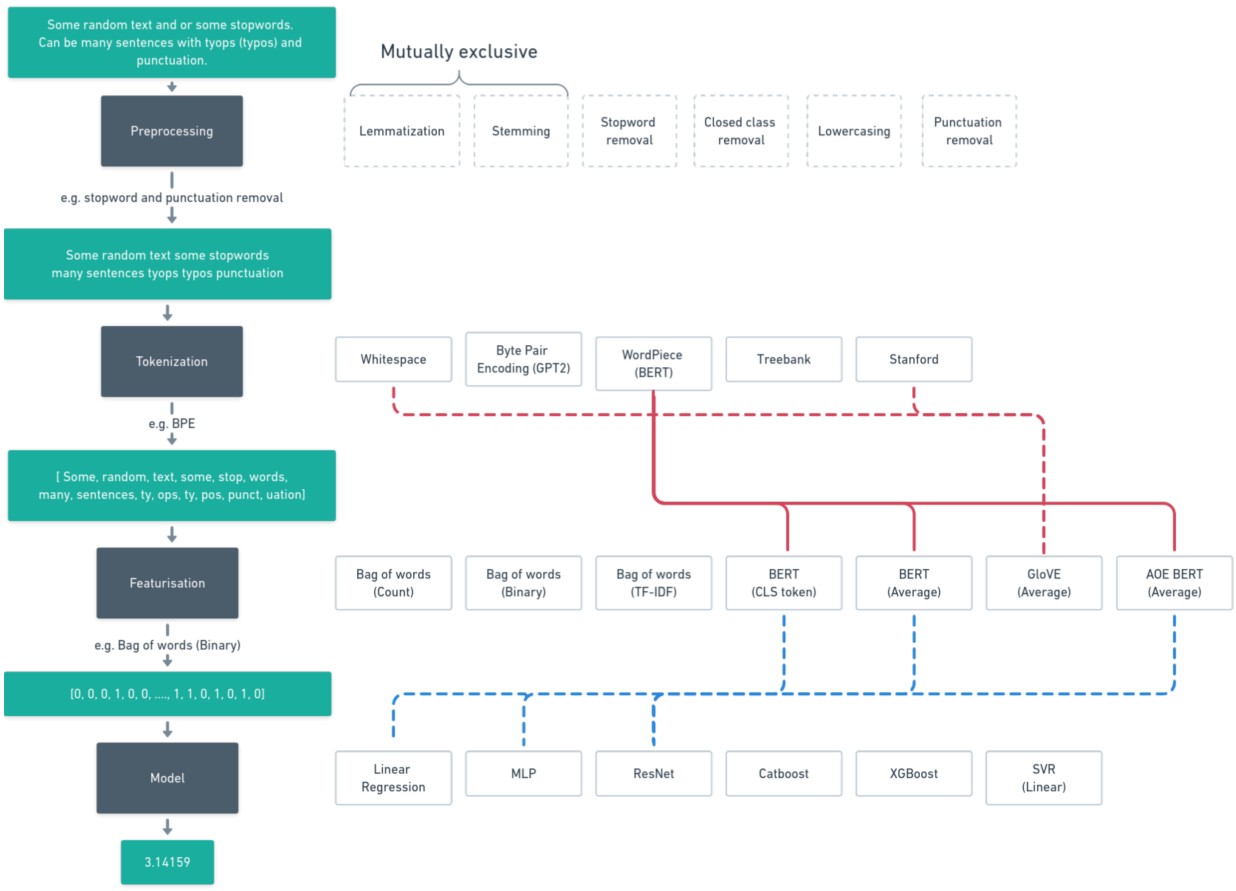

Figure 1: An NL regression pipeline with choices for each step. We show the flow of information for an example sentence beginning '*Some random text...*' in green boxes. Dashed boxes represent choices that can be combined, solid boxes are mutually exclusive. Red lines represent that the featurizer must be used with connected tokenizers. Note, a tokenizer with red line connections may be still used with other featurizers. Blue lines represent that the given modelling choice can be finetuned with the given featurizer. Dashed red lines used to separate path from solid red lines. Note: Shows new elements - AOE bert and SVR

## 2 Background

A typical NLP pipeline comprises four steps (Kuo, 2023):

- Preprocessing: maps text to text. The aim is to remove information which is irrelevant and make relevant information easier to extract. Multiple different preprocessors may be used in the same pipeline.

- Tokenization: maps text to a sequence of 'tokens', which are chunks of the text that will be used in the featurization algorithm.

- Featurization: maps a sequence of tokens to a numeric representation.

- Modeling: maps a numeric representation to a prediction.

We summarize the choices we study for each of these steps in Figure 1. A brief explanation of the choices in each step follows.

## 2.1 Preprocessing

In this section we summarize the preprocessors that we study. Notably, whilst all of these preprocessors aim to improve performance, they may have unintended consequences.

**Stemming and Lemmatization** are both methods to map different forms of the same concept to one form. For example, a document may contain the words 'ran' and 'running', or 'room' and 'rooms' and instead of treating these words as distinct concepts we may wish to map them to the same concept to enable fitting to the concept itself instead of fitting to (potentially sparse) forms. The difference between stemming and lemmatization is in how this is done.

Stemming is a heuristic process in which parts of words are deleted or altered according to a rule set, for example a rule might be the removal of 's' at the end of any word. This would map most plurals to their singular form but would map singular words ending in 's' to a non-word, e.g. 'miss' to 'mis'. In this work we include the Porter stemmer Porter (2001).

Lemmatization is a more complex process that attempts to map words to their *base form* (e.g. 'am', 'are' and 'is' are all mapped to 'be'). We include the spaCy lemmatizer Honnibal et al. (2020) in our study. The lemmatizer first predicts part of speech (e.g. verb, adjective etc.) for each word, then transforms suffixes and finally uses a lookup table in order to predict a word's lemma.

As both of these preprocessors aim to achieve the same goal, we make the choice of them mutually exclusive.

**Stopword removal** is the process of removing words that are common among all text analyzed and thus are hypothesized to have little predictive relevance. We include two approaches for stopword removal in our study that are not mutually exclusive. The first removes words that are in a predefined list as provided by the NLTK package Bird (2006). The second removes all words that are in a 'closed class' of connecting English words such as prepositions, see Dependencies for a complete list of closed class parts of speech. In short, this only leaves words that are expected to be verbs, adverbs, nouns, adjectives or numerals.

**Punctuation removal** has similar logic to stopword removal. The presence of some common punctuation in a sentence is hypothesized to have little predictive relevance (Bird, 2006) and so is removed.

**Lowercasing** maps all characters to their respective lowercase forms. For English this essentially ensures words that appear at the beginning of a sentence map to the same representation as in the middle of a sentence; as well as working around capitalization inconsistency among input texts. However, this could conflate proper names and nouns, e.g. 'President Bush' would become 'president bush' which when tokenized could map the former head of state to the same token as vegetation.

## 2.2 Tokenization

In this section we summarize the tokenizers we select for this study.

**Whitespace** tokenization defines tokens as anything separated by whitespace characters. For space-delimited languages, such as English, this is the simplest form of tokenization.

**Treebank** is a tokenizer that splits on whitespace but also uses a regex to handle some edge cases for contractions (e.g. 'don't' is tokenized to 'do' and 'n't') and punctuation.

The **Stanford** tokenizer began as an adaptation of Treebank and uses a further set of custom rules crafted by NLP researchers Manning et al. (2014).

**Byte-pair encoding (BPE)** tokenization (Sennrich et al., 2016) learns a tokenizer from text. Two key distinctions between BPE and the Whitespace, Treebank and Stanford tokenizers are i) BPE can produce sub-word tokens, e.g. 'bedroom' could be tokenized as `['bed','room']` and ii) BPE is typically pretrained from a separate body of text. We specifically include the GPT2 BPE tokenizer, that was trained as part of Radford et al. (2019).

**WordPiece** tokenization Devlin et al. (2019) is another form of trained sub-word tokenizer, with a very similar construction logic to BPE but with a different objective function during training.

### 2.3 Featurization

Given a sequence of tokens representing the input, featurizers map the sequence to a single numeric representation.

A **bag of words (BoW)** featurizer is defined by a set of tokens, $v_i \in \{v_1, v_2, ..., v_V\}$, called a vocabulary. This vocabulary of tokens should be a subset of the tokens produced by the tokenizer. The bag of words representation of some tokenized text is then a vector, $f \in \mathbb{R}^V$, where $f_i$ is the count of token $v_i$ in the tokenized text. For example, given a vocabulary `[a,b,c]` and a tokenized input `[b,b,a,a,b,d]` the bag of words representation could be $f = [2, 3, 0]^T$. Tokens that are not in the vocabulary are discarded.

BoW is order invariant, as can be seen in the prior example: swapping the positions of any `a` and `b` would not change the featurization. One way of partially mitigating this drawback is to include some *N-grams* in the vocabulary. *N*-grams are token sequences of length *N* and instead of matching a token to increase the count of the vocabulary index; the whole *N*-gram must be matched. Extending the previous example, if the vocabulary was to include a 2-gram, `[a,b,c,ba]`, the representation would be $f = [2, 3, 0, 1]^T$. In this work we choose the vocabulary to include the 50,000 most frequent tokens or 2-grams in the training set.

We also include a *modified* BoW implementation where $f_i$ is a binary indicator on the presence of $v_i$ in the tokenized text; the first example would thus be featurized as $f = [1, 1, 0]^T$. We distinguish the count and the indicator implementations through the featurizer names `bow-count` and `bow-binary` respectively. Furthermore, we include an extension of `bow-count` called `tf-idf` Salton (1991). This extension normalizes counts in a manner that down-weights counts for tokens that are common among many entries in the training set.

Alternatively, **vector embeddings** are mappings between *single* tokens and *fixed size* vectors – instead of mapping *sequences* of tokens to *vocabulary sized* vectors. These vectors are learned from very large datasets to perform well for a generic objective (such as predicting words that are masked) and then can be re-purposed for other NLP tasks Pennington et al. (2014) Mikolov et al. (2013) Devlin et al. (2019). As vector embeddings are learned mappings for *specific* tokens, vector embedding featurizers must be used with *specific* tokenizers. Otherwise, the tokens may be not present in, or semantically different to, the featurizer's vocabulary. This is one example of why not all paths through the Figure 1 are possible.

Vector token embeddings can be divided into two classes: contextual and non-contextual. Non-contextual word embeddings are purely a lookup of a token in a mapping between tokens and vectors. We include the GloVe featurizer Pennington et al. (2014) as a non-contextual word embedding. Alternatively, contextual word embeddings process a sequence of non-contextual token vector embeddings through a function that alters each vector conditional on the others. This function is usually a deep neural network. The goal of contextual vector embeddings being to have the same token, say 'bank', featurized differently in different contexts, say in finance and river geography. We include the BERT Devlin et al. (2019) featurizer as a popular Transformer based contextual word embedding in our study.

As vector embeddings produce a variable sequence of vectors, matching the sequence of tokens, we are required to pool them to a fixed size. One of the approaches we use for pooling is an elementwise mean on the token embeddings. That is, for a sequence of $T$ vector embeddings, where $T$ varies between input texts, we represent the the whole sequence as an elementwise mean across the sequence. This has the effect of mapping $T$ (variable) vectors to a one vector of size $d$ (fixed), where $d$ is the size of each individual embedding. Instead of using elementwise pooling, one could also *learn* an embedding aggregation. The BERT featurizer in fact has such an implementation, called the `CLS` 'token' output. The `CLS` output is a contextual embedding trained to capture an overall embedding for an input *sentence*. To differentiate the two methods of pooling using BERT, we call them `bert-mean` and `bert-cls` respectively.

We also include a more modern adaptation of BERT specifically designed for text embedding Li & Li (2024), for which we use mean pooling and refer to it as `aoe-bert`.

### 2.4 Modeling

Once the rest of the pipeline is chosen, we apply some machine learning model to the obtained numeric representation, solving a standard regression problem. We include Catboost Prokhorenkova et al. (2018), XGBoost Chen & Guestrin (2016), a multi layer perceptron (MLP), a residual network (ResNet), a support vector regression (SVR) with a linear kernel and a linear model trained through gradient descent as choices for the modelling step of the pipeline. The linear kernel for our SVR was chosen primarily for practical computational speed, but was also used in some prior work (Kogan et al., 2009).

We note a nuance of the modelling: if the model can be trained through gradient descent we call this model 'differentiable' (i.e. MLP, ResNet and the linear model). If the model is differentiable **and** the featurizer is also pretrained through gradient descent (i.e. a BERT model) then we can jointly train both the model and the featurizer. This is done in a process we refer to as **finetuning**. Firstly only the model is trained, with the featurizer frozen, until early stopping terminates the training. Then the model and featurizer are jointly trained with a very small learning rate until early stopping is triggered.

### 2.5 'Traditional' and 'Modern' approaches

We note that historically, two relatively independent approaches have been developed for NLP, centered around the choice of featurizer. First, a more 'traditional' framework uses featurizers that originate from a bag of words, with other components being variable. Alternatively, 'modern' deep learning approaches use featurizers that generate vector embeddings (which induce fixed tokenizer choice), minimal preprocessing, and models that allow for finetuning. Few works have studied mixing these two approaches for NL regression.

## 3 Related work

Although there is much literature on the task of NL *classification* Minaee et al. (2021), there is much less attention on NL *regression*. Prior work has mostly compared a small number of preselected pipelines or even a single pipeline with one step varied.

Early work by Joshi et al. (2010) investigated linear regression for movie box office gross from reviews. They used metadata and BoW featurization, but did not investigate the impact of altering pipeline steps. This work was followed up by Bitvai & Cohn (2015) which compared the results of Joshi et al. (2010) to a new pipeline consisting of convolutional neural networks on non-contextual vector embeddings. The work of Bitvai & Cohn (2015) in principle validated a deep learning approach to NL regression, by achieving substantially better accuracy metrics than Joshi et al. (2010) with a noticeable improvement due to the inclusion of text.

More recently, studies of NL regression focused mainly on topics related to finance, where several groups attempted to predict stock volatility using company reports. Dereli & Saraclar (2019) investigated a convolutional neural network model with a non-contextual word embedding featurizer and found an improvement over support vector regression with BoW Tsai & Wang (2014). Yeh et al. (2020) varied both the featurizer and modelling step. They compared i) TF-IDF, ii) pretrained non-contextual vector embeddings and iii) custom non-contextual vector embeddings using both linear regression and support vector regression. It was found that using a vector embedding representation achieved better accuracy than a TF-IDF approach; with a minor edge to custom vector embeddings. However, the preprocessing and tokenization of Yeh et al. (2020) was standardized across the experiments. Caron & Müller (2020) studied contextual vector embedding featurization for stock volatility regression. The study compared BoW to BERT as a text featurization strategy for support vector regression modelling. The analysis concluded that using BERT for text featurization was more accurate than BoW. Interestingly, they found, for the first time, a text-only model that could achieve better accuracy than a model using historic numeric only features. However, we note that this superiority was partly due to a large improvement in one specific year where all models generally achieved very poor accuracy.

Looking in the field of text classification, the task of proposing and comparing different featurization algorithms is a common research question. There is thus many studies comparing different approaches for

Table 1: Descriptive statistics of the response labels for the datasets studied. We note the smallest 10 responses in `obl` were under 1,000. These samples corresponded to auctions that began at very low values and test listings, the remaining listings began at 99,000.

|     | Min | Mean | Median | Max | Std |
| --- | --- | --- | --- | --- | --- |
| jcp | 4 | 128 | 35 | 1.14e+04 | 516 |
| obl | 15 | 2.48e+05 | 9.50e+04 | 1.20e+07 | 6.06e+05 |

featurization, be it from the perspective of benchmarking (Muennighoff et al., 2022; Enevoldsen et al., 2025) or in order to argue for the authors' new featurization (Devlin et al., 2019; Li & Li, 2024; Muennighoff et al., 2024; Ni et al., 2022). We note that typically these studies do not fully isolate the choice of featurization specifically; as each featurizer may have it's own tokenizer and/or classification model. Also, BoW featurization is often not included as a reference algorithm, as recent literature has focused on deep learning (DL) approaches. Although less common, recent work has began to analyse the impact of other components in the pipeline in a more isolated manner. For example, Ali et al. (2024) highlight the importance of training tokenization algorithms, especially focusing on multilingual applications. However, to the authors' knowledge, the joint impact of varying multiple pipeline components is relatively understudied even in the context of classification. The most similar work (Åvec et al., 2020) varies tokenization for a fixed featurizer and model, as well as preprocessing the data through data enrichment (e.g. suffixing a predicted part of speech tag). They find that simple whitespace tokenization can perform as well as more complex approaches; and that data enrichment preprocessing did not improve performance. Åvec et al. (2020) also study the impact of featurizer choice, although it seems that this is with each featurizer having their own preprocessing and tokenization and model. Overall, they find that a relatively simple mean pooled non-contextual vector embedding, that was trained from scratch on their data, outperformed use of contextual 'sentence embedding' featurizers. However, it was unclear how these contextual featurizers were trained or finetuned, if at all.

Overall, no prior work has systematically varied all NLP pipeline components simultaneously, leaving many potential configurations, including hybrid approaches, underexplored. For instance, whether there is merit in using vector embeddings as features for gradient boosted decision trees. In this study we aim to explore the impact of pipeline construction as a whole: allowing all parts of the pipeline to vary.

## 4 Method

A valid pipeline was generated by randomly selecting choices for each step. After random selection the choices were checked for compatibility, i.e. the BERT featurizer could not use the Treebank tokenizer. Valid pipelines are summarized in Figure 1. Initially, a random subset of preprocessors was chosen. This includes the choice of no preprocessing. If both lemmatization and stemming were chosen we randomly selected only one of them to keep in the subset. Then a tokenization, featurization and modelling strategy were also chosen at random. If the featurization strategy was only compatible with a specific tokenizer we reselected the tokenizer with a random choice from the valid tokenizers, i.e. Whitespace or Stanford tokenization for GloVe embeddings and only BERT WordPiece tokenization for `bert-cls` or `bert-mean`. Finally, if the generated model and featurizer were capable of being finetuned we randomly selected whether to finetune or not.

This was done for two real world datasets: i) a dataset for predicting consumer good prices from textual descriptions, which we call `jcp`, with 13,575 samples, and ii) a dataset for predicting boat prices from online auction textual descriptions, which we call `obl`, with 1,850 samples. The response distributions for each dataset are distinctly right skewed (characteristic of price data) and are summarized in Table 1.

Summaries of statistics of text length, and missingness, are presented in Appendix B. Samples of the datasets are also presented in Appendix B. These samples show that both datasets are related to product sales, and thus show aspects of 'keyword stuffing'. However, `obl` uses subjectively more prose-like language, whilst the online retailer `jcp` dataset uses samples that are more to the point and highlight key features.

We generated hundreds of random valid pipelines for each dataset and trained them on a 60/20/20% train/validation/test split; recording test set normalized-root mean squared error (RMSE) for each. We reshuffled the train/validation splits 4 times for each random pipeline configuration, using these reshuffled datasets for Monte Carlo cross validation. Therefore, we have 4 samples of RMSEs for each pipeline configuration. The random nature of the generated compatible pipelines was an intentional design decision, to remove subjectivity in component choice and allow for investigation of understudied combinations.

Hyperparameters were either left as defaults (e.g. default number of iterations for GBDTs) or subjectively chosen as 'typical' choices (e.g. learning rates and optimizers used for DL models). This means the analysis presented in this work is studying 'default' pipeline configurations, that are likely to be representative of common usage. For reproducibility, all experiment code, hyperparameter defaults, datasets and analysis code are made available on GitHub. We also make the results available at the same GitHub link, enabling investigation without needing to rerun simulations. The count of choices at each step across the generated pipelines is summarized in Appendix A, showing good coverage and a good number of replicates for each choice.

## 5    Results and Discussion

This section discusses interesting aspects of pipeline construction that were revealed through analysis of the results obtained from the method described in Section 4. Section 5.1 compares contextual to non-contextual vector embeddings. Section 5.2 discusses the importance of finetuning for BERT featurized methods. Section 5.3 discusses the relative performance of BoW to BERT featurization. Section 5.4 discusses the importance of choices upstream from modelling. Section 5.5 discusses the performance of 'traditional' GBDT models with 'modern' vector embeddings. For clarity, when we refer to 'upstream' choices the most upstream choice is preprocessing and the most downstream choice is model selection. All statistical test results presented within this section are from two-tailed tests. The primary metric presented is a normalized form of RMSE: where the RMSE has been divided by that which would be obtained from predicting the training data mean as a constant prediction. This denominator is analogous for regression as predicting a majority class is to imbalanced classification and is a simple baseline, allowing for easier interpretation of results. For clarity, a normalized-RMSE of 0.8 corresponds to a 20% improvement over a constant mean prediction baseline. If comparing two methods, then a mean difference of -0.07 corresponds to the former method performing 7% better (relative to a mean prediction) than the latter.

### 5.1    Contextual word embeddings outperform non-contextual word embeddings

In this section we study the relative performance of contextual and non-contextual word embeddings. We selected *pairs* of pipelines that use either BERT (-cls and -mean) or GloVe featurization. That is each pair is the same except one uses BERT featurization and the other uses GloVe (e.g. use the *same* train/validation split, preprocessor and model). This enables a pairwise comparison of featurization approaches, where the only difference is featurization. For clarity, tokenizers were not the same between these pairs, as tokenizers that are compatible with BERT and GloVe are mutually exclusive and the computational cost of retraining them was outside the scope of this study.

We then performed a paired Monte Carlo permutation test on mean normalized-RMSE between these pairs of pipelines. Table 2 shows the mean difference and p-values. The paired permutation tests shows that BERT featurization has significantly lower normalized-RMSE than GloVe featurization for both datasets. This suggests that contextual vector embeddings are better for textual regression than non-contextual embeddings, for these datasets.

### 5.2    Finetuning is required for BERT methods to outperform bag of words

In this section, we study the importance of finetuning in comparing contextual word embeddings to BoW. We selected pipelines that either used BoW featurization with any model, or used BERT featurization *with a differentiable model*. A differentiable model being one that is capable of backpropagating loss gradients for training, i.e. linear, MLP or ResNet. The BERT pipelines were then partitioned into those that did or

Table 2: Paired permutation tests for comparison of BERT and GloVe featurization strategies. 'mean-difference' reports the difference in average normalized-RMSE between BERT and GloVe respectively, i.e. negative mean-difference indicates BERT was on average more accurate. Test performed with 20 pairs for each dataset.

| dataset | mean-difference (BERT - GloVE) | p-value |
|---------|-------------------------------:|---------|
| obl     | -0.069                         | <0.001  |
| jcp     | -0.068                         | <0.001  |

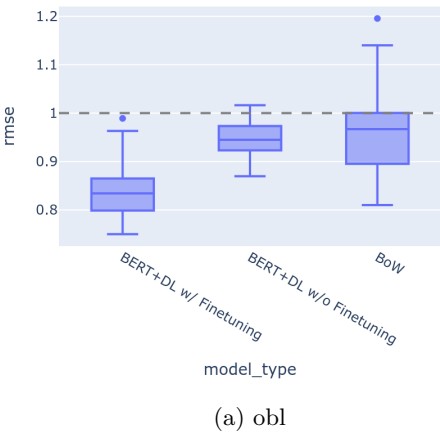
(a) obl

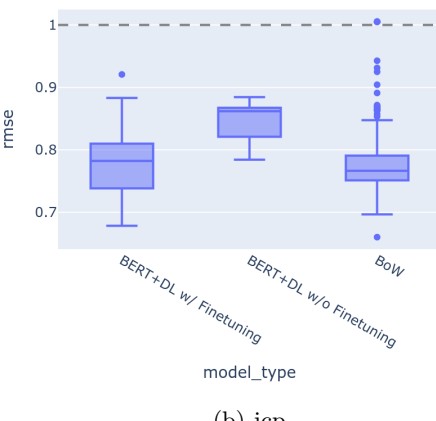
(b) jcp

Figure 2: normalized-RMSE distributions across featurization types for different datasets. The dashed line is RMSE of a baseline predicting the training set mean.

did not use finetuning. Overall, this results in three types of pipeline: BERT with finetuning, BERT with differentiable models but without finetuning and BoW pipelines. Figure 2 and Table 3 show boxplots and Monte Carlo permutation test comparing these three approaches.

The 'ft vs no-ft' rows of Table 3 show that for both datasets finetuning is significantly better than not finetuning compatible pipelines. The 'no-ft vs bow' rows of Table 3 show that models without finetuning are either comparable (`obl`) or significantly worse than BoW (`jcp`). The 'ft vs bow' row of Table 3a shows that finetuning significantly outperforms BoW for the `obl` dataset. However, the 'ft vs bow' row of Table 3b shows that BoW and finetuning are not significantly different in performance for `jcp`, with relatively small mean difference and with comparable normalized-RMSE range (shown in Figure 2b).

Overall, BERT featurization without finetuning had comparable or worse accuracy than BoW. However, with finetuning, BERT accuracy substantially improved, coming close to or significantly surpassing the performance of BoW. This suggests that although better than non-contextual GloVe embeddings, finetuning is necessary for contextual BERT embeddings to outperform BoW, for these datasets. Appendix C presents a breakdown of performance similar to Figure 2, with `aoe-bert` separated from `bert-cls` and `bert-mean`. This appendix shows that although the more modern approach (`aoe-bert`) performs better without finetuning; the vanilla implementation (`bert-cls` or `-mean`) either matches (`jcp`) or beats (`obl`) `aoe-bert` in accuracy with finetuning.

### 5.3 Bag of words approaches are still relevant for resource constrained applications

In this section we discuss the relevance of BoW featurizers in the context of time and memory constrained applications. Table 4 and 5 show the ten pipelines with lowest average normalized-RMSE, where the average is taken over Monte Carlo cross validation folds. Although we find that a finetuned BERT-featurized pipeline

Table 3: Monte Carlo permutation test for comparison of featurizations. P-values from 100,000 permutations. Negative mean difference indicates the first choice is better. 'ft' (52 samples) indicates BERT with finetuning. 'no-ft' (56 samples) indicates BERT featurizer with differentiable model but *no finetuning*. 'bow' (288 samples) indicates BoW featurizer with any model.

| comparison | mean-difference | p-value |
|---|---|---|
| ft vs no-ft | -0.105 | <0.001 |
| ft vs bow | -0.119 | <0.001 |
| no-ft vs bow | -0.013 | 0.206 |

(a) obl

| comparison | mean-difference | p-value |
|---|---|---|
| ft vs no-ft | -0.070 | <0.001 |
| ft vs bow | <0.001 | 0.908 |
| no-ft vs bow | 0.069 | <0.001 |

(b) jcp

Table 4: Top ten performing pipelines by normalized-RMSE on `obl` test set (sorted by mean, smaller is better). Standard error of mean denoted by 'sem'. Models with finetuning denoted by 'ft'. Mean and sem over 4 cross validation replicates.

| preprocesser(s) | tokenizer | featurizer | model | mean | rmse sem |
|---|---|---|---|---|---|
| empty | bert | bert-mean | deep-linear-ft | 0.786 | 0.004 |
| Le,Lo | bert | bert-mean | deep-linear-ft | 0.787 | 0.014 |
| empty | bert | bert-cls | deep-linear-ft | 0.794 | 0.019 |
| Le,Lo,NP,NSN | bert | bert-cls | deep-linear-ft | 0.822 | 0.017 |
| empty | bert | bert-cls | mlp-ft | 0.823 | 0.015 |
| Le | bpe | tf-idf | mlp | 0.824 | 0.005 |
| empty | bert | bert-mean | mlp-ft | 0.832 | 0.020 |
| Le | bpe | tf-idf | deep-linear | 0.833 | 0.005 |
| Le,NSN | bert | bert-mean | resnet-ft | 0.839 | 0.019 |
| empty | bpe | tf-idf | deep-linear | 0.841 | 0.005 |

Le = Lemmatize, Lo = Lowercase, NP = No punctuation, NSN = No stopwords (NLTK), NSCC = No stopwords (closed class), S = Stem

had the lowest average normalized-RMSE for both datasets studied, the performance of BoW models is still competitive. For `jcp` the third best pipeline uses tf-idf (a BoW featurizer) and had mean normalized-RMSE within one standard error from the best pipeline. We also note that several of the top ten pipelines for both datasets used BoW approaches. This suggests that competitive performance may be achieved using BoW, for these datasets.

The requirement for finetuning to reach comparable, or better, accuracy than BoW, discussed in Section 5.2, introduces substantial computational cost for longer text. Transformer-based featurizers, like BERT, scale quadratically with input length in memory and time due to their attention mechanisms. Furthermore, this quadratic cost is incurred for every *step* of the finetuning process, which can take multiple *epochs*. In contrast, BoW featurization scales linearly in both memory and time (assuming a fixed vocabulary size) and is trained only once. Therefore, the competitive performance of BoW suggests that BoW is a valid choice for long text inputs or when computational resources are limited. We show the difference in mean runtime between finetuned and BoW models, along with a Monte Carlo permutation test, in Table 6. This table shows that BoW pipelines are significantly faster than finetuned pipelines for regression on the datasets studied.

Table 5: Top ten performing pipelines by normalized-RMSE on `jcp` test set (sorted by mean, smaller is better). Standard error of mean denoted by 'sem'. Models with finetuning denoted by 'ft'. Mean and sem over 4 cross validation replicates.

| preprocesser(s) | tokenizer | featurizer | model | mean | rmse sem |
|---|---|---|---|---|---|
| NSN | bert | bert-mean | mlp-ft | 0.715 | 0.017 |
| empty | bert | bert-mean | mlp-ft | 0.719 | 0.011 |
| Le,Lo,NSN | stanford | tf-idf | mlp | 0.728 | 0.007 |
| empty | whitespace | tf-idf | mlp | 0.728 | 0.011 |
| NSN | whitespace | bow-count | mlp | 0.730 | 0.011 |
| empty | treebank | tf-idf | mlp | 0.731 | 0.008 |
| | bert | bert-mean | deep-linear-ft | 0.733 | 0.012 |
| | stanford | tf-idf | mlp | 0.735 | 0.009 |
| NSN,S | whitespace | glove-mean | resnet | 0.737 | 0.005 |
| Lo,NSN | treebank | bow-binary | mlp | 0.737 | 0.009 |

Le = Lemmatize, Lo = Lowercase, NP = No punctuation, NSN = No stopwords (NLTK), NSCC = No stopwords (closed class), S = Stem

Table 6: Monte Carlo permutation test for mean difference in runtime, in seconds. P-values from 100,000 permutations. Negative mean difference indicates that BoW is faster. BoW and `ft` have 288 and 52 samples respectively.

| dataset | mean-difference (BOW - FT) | p-value |
|---|---|---|
| obl | -1,372 | <0.001 |
| jcp | -3,183 | <0.001 |

Table 7: Effect of upstream modelling choices on RMSE. Maximal % reduction, from worst to best pipeline, in RMSE for different modelling strategies.

| Dataset | catboost | deep-linear | deep-linear-ft | mlp | mlp-ft | resnet | resnet-ft | svr | xgboost |
|---------|----------|-------------|----------------|-----|--------|--------|-----------|-----|---------|
| obl     | 13       | 25          | 12             | 25  | 22     | 15     | 19        | 5   | 57      |
| jcp     | 17       | 20          | 22             | 25  | 18     | 15     | 17        | 18  | 37      |

### 5.4 Pipeline steps upstream from modelling have performance impact comparable to model choice

In this section we discuss the relative importance of model selection compared to choices upstream from modelling. One way to measure this effect is to find the % reduction in normalized-RMSE from the best to the worst pipeline, conditional on model choice, across all pipelines, shown in Table 7. For example, in the obl row with the Catboost column we read 13. This indicates that across all of the pipelines which used an Catboost as the model choice, the difference between worst and best pipeline corresponds to a 13% reduction in normalized-RMSE. Table 7 shows that optimal selection of upstream steps can reduce RMSE by up to 57%. Although this measure is an approximate upper limit on the improvement for a given model (as it is measuring from worst to best), it nonetheless suggests any given model can be substantially improved by tuning of the upstream steps.

Further investigating the importance of choices upstream of modelling, the variance associated with choosing the model was compared to the variance associated with choosing the upstream steps. We use this variance to quantify how 'important' it is to optimize certain parts of the pipeline (similar to the approach of Hutter et al. (2014)), assuming larger variance implies a greater potential for improvement from optimization. The variances were quantified through empirical distributions as follows.

We defined an 'upstream group', as a group of pipelines where every pipeline only differs by model, i.e. has the same preprocessing, tokenization and featurization. We then selected every 'upstream group' with more than one modelling choice, as variance calculation requires more than one sample. Variance in mean normalized-RMSE, across modelling choices, was determined for each group and then averaged across all groups, effectively computing a variance associated with choosing the model. This average variance is denoted as $\bar{V}_{\text{model}}$:

$$\bar{V}_{\text{model}} = \frac{1}{N} \sum_i^N \text{Var}[\overline{\text{RMSE}}_i] = \frac{1}{N} \sum_i^N \frac{1}{J_i - 1} \left( \sum_j^{J_i} \left[ \overline{\text{RMSE}}_{i,j} - \frac{1}{J_i} \sum_j^{J_i} \overline{\text{RMSE}}_{i,j} \right]^2 \right), \tag{1}$$

where $i$ indexes *between* upstream groups; $N$ is the count of upstream groups; $j$ indexes *within* upstream groups which have $J_i$ different models and $\overline{\text{RMSE}}_{i,j}$ is average RMSE across cross validation folds for the $j$th pipeline in the $i$th upstream group. That is we estimate variance over *model* choices as each upstream group has fixed upstream choices.

We defined a 'model group', as a group of pipelines where each pipeline has the same model choice. For each 'model group', the variance in mean normalized-RMSE was determined. This variance was then averaged across all models, effectively computing a variance associated with choosing the upstream steps. This average variance is denoted as $\bar{V}_{\text{upstream}}$ and follows the same as Equation 1 but with $i$ indexing a *model group* instead, and $j$ indexing between different upstream pipelines. That is we estimate variance over *upstream* choices as each model group has a fixed model choice.

Table 8 presents these variances. For both datasets, the variance associated with upstream choices is of similar magnitude to that associated with the model choice. This suggests that tuning the pipeline upstream from modelling is as important as selecting the model, for these datasets.

### 5.5 Vector embeddings are not suited to regression with GBDTs

In this section we compare vector embeddings to BoW featurizers for GBDT models. Figure 3 shows the performance distribution with and without vector embedding featurization for GBDT models across both

Table 8: Comparison of average variance across either upstream pipeline or model choice.

| Dataset | $\bar{V}_{\text{model}}$ | $\bar{V}_{\text{upstream}}$ |
|---------|---------|------------|
| obl | 11e-03 | 3.0e-03 |
| jcp | 4.6e-03 | 1.7e-03 |

Table 9: Monte Carlo permutation test results for comparison of BoW and vector featurization for GBDT pipelines. P-values from 100,000 permutations. 'mean-difference' is difference of normalized-RMSE between BoW (108 samples) and vector embeddings (92 samples), i.e. negative mean difference suggests that GBDT without vector embeddings is more accurate.

| dataset | mean-difference (BoW - Vector) | p-value |
|---------|-------------------------------|---------|
| obl | -0.077 | <0.001 |
| jcp | -0.049 | <0.001 |

datasets. It can be seen that the performance of GBDTs was better for pipelines with traditional BoW featurizers (i.e. `bow-count`, `bow-binary`, `tf-idf`) instead of vector embeddings (i.e. `bert-cls`, `bert-mean`, `aoe-mean` and GloVe). Table 9 presents Monte Carlo permutation test results comparing GBDT pipelines using BoW and vector embedding featurization. We find that there is a significant difference in mean normalized-RMSE between BoW featurization and vector featurization for GBDTs. This suggests that for NL regression, the pipelines mixing 'traditional' GBDTs and 'modern' vector embeddings are less accurate than using BoW with GBDTs, for these datasets.

## 6 Conclusion

This study systematically evaluated the impact of pipeline design choices on textual regression accuracy by analyzing randomly generated pipelines on two NL regression datasets, revealing several insights.

We found that contextual vector embeddings from BERT significantly outperformed non-contextual GloVe vector embeddings, highlighting the value of capturing context-aware semantic representations. However, BERT featurization only achieved comparable (`jcp`), or substantially better (`obl`), normalized-RMSE over traditional BoW when finetuned. This emphasizes the need to adapt pretrained language models to specific applications. We also found that generally BoW approaches are capable of providing good performance for natural language regression, which could be particularly relevant for long textual inputs or with limited computational resources.

Furthermore, we found that upstream pipeline components – such as preprocessing, tokenization, and featurization – contributed significant variability in performance. This variability was comparable to model selection itself. This comparable variability highlights the importance of optimizing upstream pipeline choices, not just the choice of model.

We also found that GBDTs paired with BoW featurization consistently outperformed GBDTs using vector embedding featurization. This suggests incompatibility between tree-based models and vector embeddings for natural language regression.

These findings collectively demonstrate the importance of holistic pipeline optimization for textual regression.

### 6.1 Limitations and future work

While this study explored a variety of pipeline configurations, it did not analyze all possible steps, such as modeling approaches like convolutional neural networks or support vector regression with an RBF kernel, which have been used in some prior work. Furthermore, a deeper exploration could involve *separately pretraining* vector embedding featurizers (e.g., BERT or GloVe) with different tokenizers and preprocessors

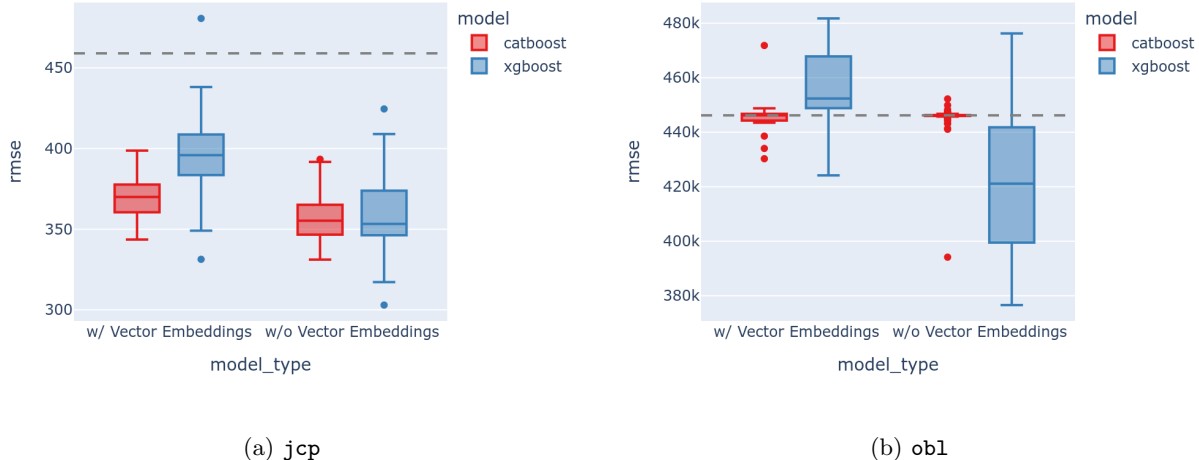

(a) `jcp`  (b) `obl`

Figure 3: Comparison of GBDT models with and without vector embeddings. The dashed line is RMSE of predicting the training set mean. For `obl` some outliers have been removed to focus on the region of interest. We also note Catboost performed poorly across all pipelines for `obl`.

from scratch, removing the practical constraints imposed by using pretrained featurizers. Although this would be computationally intensive it is an interesting avenue for future work.

Given further computational budget, it would also be interesting to study the importance of HPO in the conclusions presented, as this study only investigates reasonable defaults. However, such study would (approximately) multiply the compute required by the number of HPO steps taken - quickly increasing the cost.

The increased computational cost of any future research could be offset by selecting fewer components of interest, or disallowing certain combinations that perform poorly - e.g. preprocessing with BERT. Notably, these selections could be justified by using the data from this study - which we make public.

It is worth noting that there is substantial difference in the performance of different pipelines for each dataset. For example, contextual vectors with finetuning beat BoW for `obl`, but only achieved comparable performance for `jcp`. In addition, the top ten pipelines (by average RMSE) are distinctly different in qualitative nature between the datasets. The top ten for the `obl` dataset is primarily composed of BERT featurizers, whereas there is a mix of BoW and vector embedding approaches in the top ten for `jcp`. This suggests that there could be underlying properties of these datasets that makes them amenable to a certain form of featurization, and discovering these properties would be an interesting direction for future research. Although we have endeavoured to make the conclusions robust with replicates and statistical tests, they are derived from two datasets. As such, the conclusions may not generalize to other tasks. Future work could expand the number of datasets studied.

Finally, as LLMs become larger and more capable, future work on regression of natural language could investigate the potential for end-to-end LLM regression. That is to take the sample to be predicted, appropriately constructing a prompt around it (e.g. '`predict the cost of this item: <description>`') and then coerce the response to a numeric value. This approach, notably, would not require any training and could leverage the implicit representations learned by large, potentially proprietary, LLMs. Future work could explore many different aspects in such an approach: from prompt construction, to templating, in-context learning and dealing with how to test for data leakage (i.e. does the LLM training data include the data used for evaluation).

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

## A  Pipeline component coverage

Table 11 and Table 10 show the coverage of randomly selected choices across preprocessor, tokenizer, featurizer and model for the `obl` and `jcp` datasets respectively. That is, the tables show how much each pipeline component choice was used for a given dataset across all simulations in this work, showing a good number of replicates for each approach.

| | |
|---|---|
| no-punctuation | 204 |
| stem | 160 |
| no-stopwords-nltk | 128 |
| empty | 168 |
| lemmatize | 160 |
| lowercase | 212 |
| no-stopwords-spacy | 136 |

(a) preprocessor coverage.

| | |
|---|---|
| bert | 300 |
| whitespace | 128 |
| stanford | 100 |
| treebank | 60 |
| bpe | 40 |

(b) Tokenizer coverage.

| | |
|---|---|
| bow-binary | 108 |
| tf-idf | 96 |
| bert-cls | 88 |
| glove-mean | 88 |
| aoe-mean | 88 |
| bow-count | 84 |
| bert-mean | 76 |

(c) Featurizer coverage.

| | |
|---|---|
| mlp | 108 |
| deep-linear | 104 |
| xgboost | 100 |
| catboost | 100 |
| svr | 100 |
| resnet | 64 |
| resnet-ft | 20 |
| mlp-ft | 16 |
| deep-linear-ft | 16 |

(d) Model coverage. '-ft' corresponds to model with featurizer finetuning.

Table 10: Component coverage across random pipelines for the `obl` dataset.

| no-stopwords-nltk | 128 |
|---|---|
| empty | 168 |
| lemmatize | 160 |
| lowercase | 212 |
| no-punctuation | 204 |
| stem | 160 |
| no-stopwords-spacy | 136 |

(a) preprocessor coverage.

| bert | 300 |
|---|---|
| whitespace | 128 |
| stanford | 100 |
| treebank | 60 |
| bpe | 40 |

(b) Tokenizer coverage.

| mlp | 108 |
|---|---|
| deep-linear | 104 |
| xgboost | 100 |
| catboost | 100 |
| svr | 100 |
| resnet | 64 |
| resnet-ft | 20 |
| mlp-ft | 16 |
| deep-linear-ft | 16 |

| bow-binary | 108 |
|---|---|
| tf-idf | 96 |
| bert-cls | 88 |
| aoe-mean | 88 |
| glove-mean | 88 |
| bow-count | 84 |
| bert-mean | 76 |

(c) Featurizer coverage.

(d) Model coverage. '-ft' corresponds to model with featurizer finetuning.

Table 11: Component coverage across random pipelines for the `jcp` dataset.

## B  Dataset samples

Summaries of statistics of the text are presented in Tables 12 and 13.

Table 12: Text statistics for `obl` dataset. 1850 samples, 0.0 missing

|  | Min | Mean | Median | Max | Std |
|---|---|---|---|---|---|
| Word Count | 2 | 213 | 136 | 5811 | 328 |
| Character Count | 10 | 1350 | 862 | 36797 | 2114 |

Table 13: Text statistics for `jcp` dataset. 13575 samples, 2.8 missing

|  | Min | Mean | Median | Max | Std |
|---|---|---|---|---|---|
| Word Count | 0 | 50 | 38 | 987 | 46 |
| Character Count | 0 | 334 | 251 | 8705 | 329 |

Tables 14 and 15 show samples of the data stratified by the label value for `obl` and `jcp` respectively. Text samples are truncated to 200 characters where they exceed 200 characters.

## C  `aoe-mean` vs `og-bert`

In this Appendix we present Figure 4. This figure shows that although the more modern approach (`aoe-bert`) performs better without finetuning; the vanilla implementation (`bert`) either matches (`jcp`) or beats (`obl`) `aoe-bert` in accuracy.

Table 14: Stratified samples from `jcp`. Price column includes the stratified percentiles that the sample was drawn from, i.e. (0-20) indicates between the 0 and 20th percentile.

| Text | Price |
| --- | --- |
| This fast-absorbing blend of oils moisturizes and nourishes the hair and scalp with essential nutrients for silky, smooth, healthy-looking hair. includes tea tree oil, peppermint oil, silk and chamomi... | $15 (0 - 20) |
| This basketweave-style tie from Van Heusen® catches the light in all your favorite colors. 3½" wide silk dry clean imported | $30 (20 - 40) |
| With a classic peasant shape, our fit-and-flare dress embraces an iconic silhouette for a bold bohemian look. 34" length from shoulder knit polyester/spandex washable made in America | $33 (40 - 60) |
| Keep him comfortable all day in our modern boys running shoes. technology flexible and lightweight Phylon promotes a natural stride construction mesh/synthetic upper synthetic/rubber sole de... | $47 (60 - 80) |
| Round and round, these earrings sparkle brilliantly with 1/3 ct. t.w. diamonds in a timeless 10K white gold setting. Metal: Rhodium-plated 10K white gold Stones: 1/3 ct. t.w. round diamonds Color: I-J Cla... | $906 (80 - 100) |

Table 15: Stratified samples from `obl`. Price column includes the stratified percentiles that the sample was drawn from, i.e. (0-20) indicates between the 0 and 20th percentile.

| Text | Price |
| --- | --- |
| Designed by the famous S&S studio in 1969, built by Winfield & Partners, it was produced in England and Australia in 400 units. Exceptional construction and level of finish, the S&S 34 is a small eleg... | $25,000 (0 - 20) |
| Well built ship. A timeless classic with many innovations and kept up to date. In 2012, the engine was overhauled, new batteries, excellent navigation equipment and a beautiful sailing wardrobe. A... | $36,000 (20 - 40) |
| Bow thruster, shore power, boiler, mast lowering installation, etc. Surprisingly spacious Blom Lemsteraak. General Hull shape: Round bottom Controls: Helmet wood Windows: In brass and in teak rebate... | $79,900 (40 - 60) |
| Make: Bénéteau Model: Beneteau 58 2010 Beneteau 58 Year: 2010 Show more | $297,000 (60 - 80) |
| THE CNB 76 FIFTY FIFTY is a sturdy, high-performance blue cruiser, featuring an aluminium hull with teak deck and Sparcraft aluminium mast and boom. The vessel has been carefully maintained and upgrad... | $690,000 (80 - 100) |

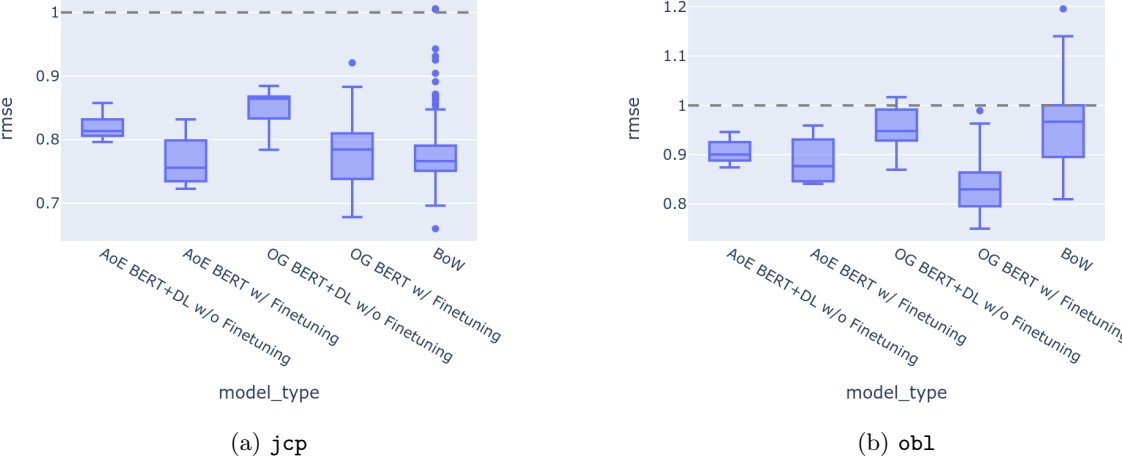

(a) `jcp`                                         (b) `obl`

Figure 4: Comparison of GBDT models with and without vector embeddings. The dashed line is normalized RMSE of predicting the training set mean.

