# OpenReview forum: "Exploring NLP pipelines for textual regression of prices"
_TMLR — Rejected by TMLR_

### Review · Reviewer_yETR · 2025-04-11

**Summary Of Contributions:**

The paper presents a systematic study of design choices across different stages of text regression pipelines. By evaluating a diverse set of randomly generated valid pipelines on two pricing-related text regression datasets, the authors derive five key findings. These insights offer practical guidance for both researchers and practitioners working on natural language regression tasks.

**Audience:**

Yes

**Broader Impact Concerns:**

No concerns.

**Claims And Evidence:**

Yes

**Requested Changes:**

* C1: Include recent state-of-the-art text embedding models to strengthen the empirical comparison (see W1).
* C2: Expand the evaluation to include datasets from diverse domains and with varied prediction targets, beyond pricing tasks (see W2).
* C3: Conduct experiments across a range of dataset sizes to better understand performance under different data availability conditions (see W2).
* C4: Reconsider the experimental methodology for certain claims, especially where technical mismatches in pipeline configurations may affect fairness or validity (see W3).
* C5: Either include additional regression models, such as SVR and linear regression, or provide a clear rationale for their exclusion (see W4).
* C6: Refine the writing to focus on key insights and reduce overly detailed explanations of well-known concepts (see W5).

Please refer to the corresponding weaknesses above for further justification.

**Strengths And Weaknesses:**

Strengths:

* S1: The paper presents several findings that are potentially insightful for the community, e.g. the result that bag-of-words (BoW) remains competitive in certain settings is an interesting empirical contribution.

* S2: The systematic investigation of NLP pipelines is particularly useful for practitioners who may lack extensive experience in textual regression.

Weaknesses:

* W1: The most significant limitation is the restricted set of models considered. Recent benchmarks such as MTEB [1] have demonstrated that plain BERT embeddings are relatively weak, often appearing near the bottom of leaderboards. Although MTEB does not cover regression tasks, its findings in classification suggest that other high-performing models should be included. The inclusion of recent strong encoder-based (e.g., [2]) and decoder-based (e.g., [3]) embedding models would provide a more comprehensive comparison. Moreover, many practitioners rely on proprietary models (e.g., OpenAI embeddings) due to limited hardware access, and such baselines would enhance the paper’s practical relevance.

* W2: The evaluation is limited to two datasets, both related to pricing. However, text regression spans a wide range of domains, such as review rating prediction [4] and student essay scoring [5]. A broader dataset selection would better support the generalizability of the findings. Additionally, key dataset statistics (e.g., sample sizes, label distributions) are missing. This information is crucial for understanding the experimental setup, especially when training set sizes may vary drastically (e.g., 10, 100, 1000, 1M samples), which can significantly affect model performance.

* W3: The methodology of constructing random valid pipelines may lead to comparisons that are not technically sound. For instance, BERT-based models are typically trained on raw text and may not benefit from preprocessing, whereas BoW-based models are more sensitive to tokenization and preprocessing. Reporting the best result from valid pipelines (rather than the mean across all configurations) may yield more meaningful comparisons.

* W4: The regression model choices lack sufficient justification. Established baselines such as Support Vector Regression (SVR) and linear regression are notably absent, despite being commonly used in regression models.

* W5: In terms of writing, the authors should prioritize discussing the most impactful insights. Some sections, particularly Section 2.1 on preprocessing, delve into textbook-level explanations that could be shortened and referenced instead, allowing space to focus on novel contributions.


References:

[1] Muennighoff, Niklas, et al. "MTEB: Massive Text Embedding Benchmark." Proceedings of the 17th Conference of the European Chapter of the Association for Computational Linguistics. 2023.

[2] Li, Xianming, and Jing Li. "AoE: Angle-optimized embeddings for semantic textual similarity." Proceedings of the 62nd Annual Meeting of the Association for Computational Linguistics (Volume 1: Long Papers). 2024.

[3] Muennighoff, Niklas, et al. "Generative representational instruction tuning." ICLR 2024 Workshop: How Far Are We From AGI. 2024.

[4] Ganu, Gayatree, Noemie Elhadad, and Amélie Marian. "Beyond the stars: Improving rating predictions using review text content." WebDB. Vol. 9. 2009.

[5] https://www.kaggle.com/competitions/learning-agency-lab-automated-essay-scoring-2/data

---

> ### Author Response · Authors · 2025-06-16
> **Response to yETR**
>
> We thank the reviewer for their detailed review. We were pleased to read the reviewer appreciated our efforts to provide practical empirical guidance to researchers.
>
> ## C1: Include recent state-of-the-art text embedding models to strengthen the empirical comparison (see W1).
> We discuss inclusion of more featurizers in the general comment. We include AoE-BERT [1] as a recent state-of-the-art adaptation of BERT.
>
> ## C2: Expand the evaluation to include datasets from diverse domains and with varied prediction targets, beyond pricing tasks (see W2).
> We discuss inclusion of more datasets in the general comment. Given the timeframe of the rebuttal period, we have opted to not extend the number of datasets at this time, instead focusing the paper on price regression and highlighting this limitation in **Section 6.1**.
>
> ## C3: Conduct experiments across a range of dataset sizes to better understand performance under different data availability conditions (see W2).
> As an extension of C2, we discuss more datasets in the general comment. However, we note that we now include more clear dataset descriptions - and highlight that the existing datasets obl and jcp are already of different sizes (1850 and 13575 samples respectively) in **Section 4**. We also add more detailed descriptive label statistics in the form of Table 1 (**Section 4**) and text length statistic in **Tables 12 and 13 of Appendix B**.
>
> ## C4: Reconsider the experimental methodology for certain claims, especially where technical mismatches in pipeline configurations may affect fairness or validity (see W3).
> > W3: The methodology of constructing random valid pipelines may lead to comparisons that are not technically sound. For instance, BERT-based models are typically trained on raw text and may not benefit from preprocessing, whereas BoW-based models are more sensitive to tokenization and preprocessing. Reporting the best result from valid pipelines (rather than the mean across all configurations) may yield more meaningful comparisons.
>
> We intentionally chose a random construction approach; however we did make every effort to reduce _invalid_ or technically _wrong_ pipelines. We believe it is exactly the inclusion of typically un-included pipelines that contrbutes to part of the novelty of our work. We vary all stages of the pipeline jointly; allowing us to more rigorously investigate questions and assumptions (for example our results could be analysed to make the statement “BERT-based models [...] may not benefit from preprocessing” and “BoW-based models are more sensitive to tokenization and preprocessing” more rigorous).
> In response to the suggestion of studying the best pipelines: **Section 5.3** does indeed discuss the best 10 pipelines for each datasets - the results of which we use to contextualise a discussion on the surprising relevance of BoW approaches.
>
> ## C5: Either include additional regression models, such as SVR and linear regression, or provide a clear rationale for their exclusion (see W4).
> We agree the choice of regression models is not exhaustive, as we originally highlighted in the conclusion.
> However, we already include a form of linear regression; except it is trained by gradient descent using MSE under early stopping. This is done to ease implementation of finetuning for compatible featurizers. The justification for this choice being that as the loss for a linear model is convex, solution via gradient descent and early stopping can be interpreted as a regularised form of a more “typical” OLS approach.
> Notably, we did in fact tried an unregularized OLS approach (not reported but present in our published data) and found the performance to typically grossly overfit and pollute results with outliers in performance. Therefore, to use a linear model we would need to use some form of regularisation - so we believe it is reasonable to use an approach that most easily integrates with other steps in the pipeline, giving the best possible performance for a linear model.
> Whilst we do not add an OLS analysis to the paper; we do agree that SVR could be interesting to include (as discussed in the original conclusion) and therefore add a linear kernel SVR to the set of models considered, shown in Figure 1 and discussed in **Section 2.4**.
>
> ## C6: Refine the writing to focus on key insights and reduce overly detailed explanations of well-known concepts (see W5).
> We believe clear writing and being accessible to a wide audience of readers is important for not only making the work self contained but to broaden the audience which may benefit from our work. Therefore, we opt to retain the level of detail in our background.
>
>
>
> [1] Li, Xianming, and Jing Li. "AoE: Angle-optimized embeddings for  semantic textual similarity." Proceedings of the 62nd Annual Meeting of  the Association for Computational Linguistics (Volume 1: Long Papers).  2024.

---

### Review · Reviewer_NbcV · 2025-04-11

**Summary Of Contributions:**

This paper provides a comprehensive empirical study of design decisions in building NLP pipelines for text-based regression tasks, in which text is used as features to predict some numerical target.

The authors systematically evaluate different decisions in four stages of an NLP pipeline: pre-processing, tokenization, featurization, and modeling. By analyzing the effect of each design decision on a large number of randomly selected pipelines, the paper generates actionable findings about embedding choices, fine-tuning, and more stages of the NLP pipeline.

**Audience:**

Yes

**Claims And Evidence:**

No

**Requested Changes:**

What would improve this paper is assurance that the findings are general. Specifically, I need to be further convinced that the results are not due to dataset selection or pipeline sampling variation. My hope is that my main points of feedback are in the service of improving the generality of the findings.

Please address the following, which map to the weaknesses outlined above. I've highlighted the three most important to me below, which are critical for me to recommend acceptance.
1. **[High importance]** What are some of the structural differences between the two datasets under consideration? Similarly, couldn't review datasets be reframed as regression tasks as well (e.g., the [plethora of review tasks on HF](https://huggingface.co/datasets?sort=trending&search=review)), or sentiment intensity prediction (e.g., SemEval Task 7, [1])?
2. **[Very high importance]** While I understand that exploring the entire space of design decisions is combinatorially infeasible, could the authors consider trying more controlled experiments (e.g., all else equal, vary one aspect)? For example, for Section 5.1, one could randomly select K different pipelines, and for each one, set the featurization strategy. Any sampled pipeline incompatible with *any* of the featurization options is also eliminated from consideration. This ensures that we've basically run an RCT with respect to the design decision of interest: i.e., for every pipeline with BERT featurization, there exists a pipeline identical in every way but for GloVe featurization. As I understand it, the paper currently selects K random pipelines, then selects ones that happen to have BERT featurization vs. GloVE featurization. Please correct me if I've misunderstood the process (and clarify the writing accordingly); the same comment applies to the other subsections of Section 5. Section 5.4 seems to already implement this suggestion.
3. How should I interpret the results/any anchor to what a "good" value would be? I wonder if a Cohen's d-style metric that controls for variance in the variable of interest is possible here.
4. **[High importance]** I think the literature review should be significantly expanded. For example, could the authors point to recent conference proceedings/influential papers and highlight how they only focus on one part of the pipeline? I would also recommend citing previous studies on the effect of each of these design decisions.
5. Could the authors speak to the relevance of their pipeline in comparison to LLMs? For example, is the pipeline under investigation appealing in resource-constrained settings? Is it more interpretable? Informally, when might one prefer to use this pipeline over an LLM? This is hinted at in 5.3, but I would love to hear this discussed more up-front.

[1] S Kiritchenko, S Mohammad, Salameh, M.  SemEval-2016 Task 7: Determining Sentiment Intensity of English and Arabic Phrases. Proceedings of the 10th International Workshop on Semantic Evaluation (SemEval-2016), 2016.

**Strengths And Weaknesses:**

## Strengths
1. The paper helps the community pinpoint which design decisions matter in an NLP pipeline and distills its insights into actionable takeaways (Section 5 subsection headers are a great example)
2. While many other works focus on optimizing one small part of this pipeline, this paper explicitly considers all components jointly.

## Weaknesses
1. A model is only as good as the data, and with only two datasets under evaluation, I'm concerned about whether the findings are highly dataset dependent.
2. The pipeline generation process is random, making it unclear whether the effect of a particular design decision can actually be estimated. For example, in Section 5.1: "We selected pipelines that use either BERT (-cls and -mean) or GloVe featurization." However, the other components of this pipeline are random and not necessarily matched, so random variation between the which components were randomly selected for pipelines with BERT/GloVe featurization could induce confounding. The same comment applies to all other experiments. Note that the total # of pipelines generated should be reported in the body as well.
3. The permutation tests report differences in RMSE across design decisions. However, the dataset target variables are likely on very different scales (i.e., boats are much more expensive than consumer goods), making it difficult to interpret the findings.
4. The literature review can be improved. The claim "no prior work has systematically varied all NLP pipeline components simultaneously, leaving many potential configurations, including hybrid approaches, underexplored" (Section 3, Related Work) is certainly believable, but needs to be better substantiated.
5. The relevance of the paper given the quality of LLMs is unclear, but this is minor: some may argue that the increasing context length of "modern" LLMs replaces the need to explicitly pre-process, tokenize, and featurize: the raw text data can be fed into a prompt and the resulting response can be readily parsed to solve a regression task.

---

> ### Author Response · Authors · 2025-06-16
> **Response to NbcV**
>
> We thank the reviewer for their time, engaging with the work and providing a detailed review.
>
> ## C1: [High importance] What are some of the structural differences between the two datasets under consideration? Similarly, couldn't review datasets be reframed as regression tasks as well (e.g., the plethora of review tasks on HF), or sentiment intensity prediction (e.g., SemEval Task 7, [1])?
>
> We discuss inclusion of more datasets in the general comment. Given the timeframe of the rebuttal period, we have opted to not extend the number of datasets at this time; choosing to focus the manuscript on price regression.
>
> Specifically on discussion of the differences between datasets: we add more descriptive statistics not only on the response (Section 4) but also on the distribution of words and overall length of the samples in each dataset (**Appendix B**). We also add a qualitative comment to **Section 4** briefly describing some differences.
>
> ## C2: [Very high importance] While I understand that exploring the entire space of design decisions is combinatorially infeasible, could the authors consider trying more controlled experiments (e.g., all else equal, vary one aspect)? For example, for Section 5.1, one could randomly select K different pipelines, and for each one, set the featurization strategy. Any sampled pipeline incompatible with any of the featurization options is also eliminated from consideration. This ensures that we've basically run an RCT with respect to the design decision of interest: i.e., for every pipeline with BERT featurization, there exists a pipeline identical in every way but for GloVe featurization. As I understand it, the paper currently selects K random pipelines, then selects ones that happen to have BERT featurization vs. GloVE featurization. Please correct me if I've misunderstood the process (and clarify the writing accordingly); the same comment applies to the other subsections of Section 5. Section 5.4 seems to already implement this suggestion.
>
> We agree with the reviewer that a more “RCT-like” approach is easier to interpret; and that finding pairwise comparisons improves the interpretability of the work. We have attempted to use this approach in subsections where we believe it to be applicable. Sections 5.1 and 5.4 both use an approach as the reviewer describes. We apologise that the original explanation in 5.1 was not clear ("Then these pipelines were further refined to pairs of pipelines, where each pair is the same except one uses BERT featurization and the other uses GloVe") and **have updated the Section 5.1 accordingly**.
> Sections 5.2, 5.3 and 5.5 do _not_ use such an approach as there is not a clear definition of the pairs (e.g. in Section 5.2 BoW and DL w/finetuning are _groups_ of featurizer, i.e. more than one algorithm) - thus it is less clear what to compare when presenting a “paired” difference.
>
> One could potentially find a pipeline shared by members of each group of featurizer, but with the same preprocessing and modelling, and then somehow compare these two sets of pipelines - although it is not immediately clear which subsets in each selection to compare and how.
> Furthermore, if the sets for comparison are not the same across paired set selections it is unclear how this affects the analysis.
> In this context we have opted for a broader “main effects” approach in sections 5.2, 5.3 and 5.5 as it is unclear what a paired analysis would be for our analysis of pipelines. Furthermore, there is the significant issue of blocking in the choices of pipeline components. For example, it is not feasible to compare BERT with BoW in an RCT manner as it would require retraining of BERT under a wider set of tokenizers - which is computationally expensive.
>
> If it is a more compelling argument for the reviewer: the reviewer essentially advocates for paired analysis in a design/analysis of experiments context - which is known to be more powerful. However, it is still common scientific practice to perform un-paired analysis to determine main effects; given the understanding that aliasing/confounding of results may reduce their robustness and we will have less power to discern real effects.
>
> ## C3: How should I interpret the results/any anchor to what a "good" value would be? I wonder if a Cohen's d-style metric that controls for variance in the variable of interest is possible here.
>
> We discuss normalization of results in the general comment. We divide the reported RMSE by the RMSE obtained by using a training set mean as a constant prediction.

---

> ### Author Response · Authors · 2025-06-16
> **Response to NbcV (2)**
>
> ## C4: [High importance] I think the literature review should be significantly expanded. For example, could the authors point to recent conference proceedings/influential papers and highlight how they only focus on one part of the pipeline? I would also recommend citing previous studies on the effect of each of these design decisions.
>
> Whilst it is naturally difficult to provide an absence of evidence, let alone an evidence of absence, **we extend the review in Section 3** to include some examples of (quite recent) work that addresses the effects of tokenization and preprocessing in isolation. Work studying the impacts of featurization is relatively standard; but often not in isolation (i.e. a whole pipeline proposed by one author is compared to that of another). Finally, as regression on text is itself a relatively new application there is few, if any, studies of the influence of any components (in isolation or not) applied specifically to the task of regression.
>
> ## C5: Could the authors speak to the relevance of their pipeline in comparison to LLMs? For example, is the pipeline under investigation appealing in resource-constrained settings? Is it more interpretable? Informally, when might one prefer to use this pipeline over an LLM? This is hinted at in 5.3, but I would love to hear this discussed more up-front.
>
> We add a sentence on the potential to use LLMs for regression directly (not through applying models to the embedding space, but rather parsing results from a prompt) to **Section 6.1**. This is an interesting new avenue for research; but we believe in order to make this paper more focused on the problem we study, it is best left to a brief mention.

---

### Review · Reviewer_NUE6 · 2025-05-25

**Summary Of Contributions:**

In this work the authors perform a systematic analysis of traditional methods for NLP regression. They include evaluations on two datasets and a wide variety of techniques - including BoW, GloVe, and BERT featurizations, as well as a large number of settings of potential hyperparameters around the overall pipeline. From this analysis the authors conclude that deep model contextualization (BERT) outperforms GloVe, but requires fine-tuning to outperform BoW, which also works better when using GBDT models. Moreover, choices involving other aspects of the pipeline can have impacts on performance equivalent to selecting the model itself. These findings essentially echo findings from the NLP community over the last few years, and thus this paper should be evaluated on the extent to which it provides rigorous, large-scale, quantifiable evidence for these conclusions.

**Audience:**

No

**Claims And Evidence:**

No

**Requested Changes:**

### Critical
1. **Additional datasets.** In order to provide a comprehensive empirical overview of methods for the task of text regression 4-6 datasets covering very different semantic areas and target types should be considered.
2. **Same pipeline configurations for all datasets.** Using the same set of pipeline samples for each dataset makes it easier to draw cross-dataset comparisons.
3. **Include the number of samples supporting a given row in a table.** While marginal distributions for sampling parameters are provided in the appendix, it would be helpful to include the number of supporting experiments for a given row in each table.
4. **Normalize evaluation metrics.** Using raw RMSE makes cross-dataset comparisons challenging.
5. **Align distribution over configurations with real-world practice.** Prune out choices which are known to be less useful. This will also reduce the number of necessary experiments, potentially allowing for greater coverage and stronger conclusions.
6. **Hyperparameter tuning details.** Include details about how you tune hyperparameters.

### Suggestions
1. **Configuration reduction.** Further reduce the number of required configuration comparisons by running small-scale experiments and modeling the importance of each factor using fANOVA [(Hutter et al. 2014)](https://proceedings.mlr.press/v32/hutter14.html). Analysis of the fANOVA model can allow you to justifiably prune various configurations before running a larger scale analysis, and hopefully allow to densely cover the remaining configuration space.
2. **Use Bayesian hyperparameter tuning.** Rigorously justify your hyperparameter tuning process by using Bayesian optimization.
3. **Include modern approaches.** Specifically, consider using a large pretrained text-embedding model with optional parameter-efficient fine-tuning.

**Strengths And Weaknesses:**

### Strengths
1. **Clear writing.** The paper is well-written, and provides a detailed overview of the techniques covered. Even someone with a limited background to these techniques should be able to read the paper and understand the author's approach.
2. **Large breadth of choices for NLP pipeline.** The authors attempt to capture a large variety of options in the pipeline, including preprocessing, tokenization, featurization, and model types, and resulting in hundreds of valid pipeline configurations.
3. **Rigorous statistical testing.** The use of matched pairs and application of paired Monte Carlo permutation tests provides rigorous statistical support to the author's conclusions.


### Weaknesses
1. **Limited dataset diversity.** As mentioned in the summary, the primary contribution of this work is to provide a systematic large-scale analysis. While the authors have attempted to run a large number of potential configurations, they have done so over just two price-prediction datasets. If the authors goal is to provide a comprehensive analysis of pipelines for NLP regression, it seems necessary to include a wider variety of datasets.
2. **Difficulty in cross-dataset comparison.** The appendix indicates that the sampled pipelines for each dataset were different. This seems unnecessary and presents direct cross-dataset comparison. In addition, the price distribution for these datasets is vastly different, making raw RMSE differences impossible to interpret. Normalizing the targets or reporting unitless metrics would be preferable.
3. **Distribution of "valid" pipelines does not align with pipelines in practice.** While the authors attempt to draw general conclusions, their method of analysis assumes the distribution over valid configurations is representative of the sort of methods which would be used in practice. For example, for the sake of argument, assume there is some configuration where GloVe outperforms BERT. We could easily change the configuration pipeline with the inclusion of a variable which actually doesn't impact the output of either model, but causes this particular configuration to be upsampled to the point where Table 1 shows the opposite conclusion. As it stands, the distribution of valid configurations presented in this paper is not very representative of what machine learning practitioners would do in practice. For example, preprocessing with BERT is rarely done, and likely not worthwhile. Similarly, using a tokenizer like BPE with BoW / TF-IDF vectorization is unusual, to say the least, and the fact that these appear in the top 10 makes me question the extent of tuning the more sensible combinations underwent.
4. **Hyperparameter tuning details.** Many of the proposed aspects of the pipeline have a variety of hyperparameters, and tuning these hyperparameters can be challenging. Incorrectly tuned methods can lead to spurious conclusions, and the authors did not include any details as to their method of rigorously tuning such a large number of models.
5. **Old methods.** In my experience, none of these methods presented would be considered today for a text regression task, largely for reasons that were identified by the authors. BERT models would not be a bad choice, however even in the original BERT paper the authors identified that it was necessary to fine-tune the CLS token to be useful. The standard approach today would be to use a large-scale pretrained text embedding model for featurization, and optionally fine-tune using a parameter-efficient fine-tuning method.

---

> ### Author Response · Authors · 2025-06-16
> **Response to NUE6**
>
> We thank the reviewer for their clear and detailed feedback. We especially liked to see that the reviewer appreciated our efforts to make the work accessible to a wide audience, as we believe making science accessible is important; especially in light of the TMLR's mission guidelines on clear writing.
> Furthermore, we agree with the framework on which the reviewer was evaluating: “evaluated on the extent to which it provides rigorous, large-scale, quantifiable evidence for these conclusions”.
> We believe by studying NL price regression we naturally access a more quantifiable measure of performance (than, say, generative NLP). We also believe use of statistical methods and including replicates in the study design demonstrates its rigour. Given the relatively new nature of the problem, and lack of established benchmarks, there is admittedly fewer datasets than one may wish for; and the “large-scale” nature of our study could be improved in that dimension. However, we note our attempt to remove bias through random generation of pipelines, and the number of sampled pipelines, hopefully improves the robustness of our results on the datasets studied.
>
>
> ## C1: Additional datasets. In order to provide a comprehensive empirical overview of methods for the task of text regression 4-6 datasets covering very different semantic areas and target types should be considered.
> We discuss inclusion of more datasets in the general comment. We have opted to not extend the set of datasets at this time.
>
> ## C2: Same pipeline configurations for all datasets. Using the same set of pipeline samples for each dataset makes it easier to draw cross-dataset comparisons.
> We have updated the set of experiments so that the same experiments/pipelines are analysed for both datasets - as can be seen from the consistency of Tables 10 and 11 in **Appendix A**. This alignment does not change any conclusions.
>
> ## C3: Include the number of samples supporting a given row in a table. While marginal distributions for sampling parameters are provided in the appendix, it would be helpful to include the number of supporting experiments for a given row in each table.
> We have updated table captions for **Tables 2,3,6 and 9** to indicate the number of supporting experiments for each analysis.
>
> ## C4: Normalize evaluation metrics. Using raw RMSE makes cross-dataset comparisons challenging.
> We discuss normalization of results in the general comment. We now divide the reported metrics by the RMSE obtained by using a training set mean as a constant prediction.

---

> ### Author Response · Authors · 2025-06-16
> **Response to NUE6 (2)**
>
> ## C5: Align distribution over configurations with real-world practice. Prune out choices which are known to be less useful. This will also reduce the number of necessary experiments, potentially allowing for greater coverage and stronger conclusions.
> We agree that identifying and removing choices which are “known to be less useful” would present an interesting analysis with a lens that may be more closely aligned to current practices. However, use of randomness could be considered a strength as it reduces the subjectivity in selection of pipelines for inclusion in the analysis.
> Furthermore, as most prior research studied comparison of pipelines overall, the “known” in “known to be less useful” lacks quantifiable evidence. In a sense, through further analysis of the results in this study one could provide the quantitative, and rigorous, evidence required to back up statements such as “known to be less useful”.
> As such, we believe our attempted “uniform” sampling (naturally imperfect as it is affected by blocking) is a reasonable distribution of pipelines a-priori to quantitative evidence.
>
> More specifically, discussing W3, which this RfC is built upon:
> > For example, for the sake of argument, assume there is some configuration where GloVe outperforms BERT. We could easily change the configuration pipeline with the inclusion of a variable which actually doesn't impact the output of either model, but causes this particular configuration to be upsampled to the point where Table 1 shows the opposite conclusion.
>
> We agree with the reviewer that random oversampling could give misleading results.
> However, our (approximately) uniform sampling reduces the chance of oversampling good GloVE and undersampling good BERT (for example) - by chance.
>
> Notably, this same argument may apply exactly to the reviewer's proposed methodology. For example, choosing the pipelines manually based on subjective beliefs may oversample previously studied combinations and give rise to misleading conclusions. However, with manual selection we do not have the benefits of random sampling (in reducing the chance that we oversample a bad configuration) and thus do not benefit from a more objective comparison.
>
> > As it stands, the distribution of valid configurations presented in this paper is not very representative of what machine learning practitioners would do in practice. For example, preprocessing with BERT is rarely done, and likely not worthwhile. Similarly, using a tokenizer like BPE with BoW / TF-IDF vectorization is unusual, to say the least, and the fact that these appear in the top 10 makes me question the extent of tuning the more sensible combinations underwent.
>
> We agree that there is the inclusion of unusual pipelines, and we consider this to be part of the novelty of the work. We attempt to avoid subjective selections in our choice of pipeline components, and thus can draw insight on atypical unstudied combinations. For example, our conclusions on the weakness of vector embeddings for GBDTs.
>
> Importantly, we believe that choices should not be pruned based on assumptions, such as a choice being ‘likely not worthwhile’ or ‘unusual’. Instead, such decisions should be grounded in quantitative evidence, which we have begun exploring in this paper.
> We also note that although “unusual”, the BPE tokenizer does indeed perform well with BoW approaches (as evidenced by its presence in the best performing pipelines). As noted by the reviewer themselves, this is a surprising result, not previously noticed in the literature that can open avenues of new research. This is an example of something we would not observe if we pruned our choices to align with our subjective notions of reasonable.
>
> We address tuning in our reply to C6.
>
> ## C6: Hyperparameter tuning details. Include details about how you tune hyperparameters.
> We do not perform any tuning of hyperparameters; using defaults where libraries provide them and “typical” choices for parameters like learning rate, which are all on our public GitHub repository. We have added a clarification of this to **Section 4**. This clarification highlights that our study allows analysis of out-of-the-box performance, a realistic and practically relevant scenario. We also add a note that inclusion of a nested HPO would drastically increase the computational cost of the experiments, but would be an interesting direction for future research, in **Section 6.1**.

---

> ### Author Response · Authors · 2025-06-16
> **Response to NUE6 (3)**
>
> ## S1: Configuration reduction. Further reduce the number of required configuration comparisons by running small-scale experiments and modelling the importance of each factor using fANOVA (Hutter et al. 2014). Analysis of the fANOVA model can allow you to justifiably prune various configurations before running a larger scale analysis, and hopefully allow to densely cover the remaining configuration space.
>
> Our reply to C5 concerning pruning configurations alludes to the implied point of this suggestion: that in order to do the pruning one must rigorously justify it.
> The question of identifying important hyperparameters generally is essentially the same as identifying important components in an NLP pipeline, and we appreciate the reviewer for bringing to light an interesting and relevant piece of literature. However, we will not use the suggested fANOVA procedure to justify pipeline generation. In some sense - the analysis presented in section 5.4 is in fact already a partitioning of variance, however instead of using estimates via trees we simply use the empirical distributions from our sampling procedure.
>
> If we understand the reviewer correctly: the reviewer suggests conditioning this on more refined subspaces of components. This refined subspace would be chosen by studying the main effects of certain featurizers and interactions between featurizers and other components via fANOVA. This would indeed be interesting, however we opt to leave it for future work. We believe the broader `unconditioned' pipeline generation procedure presented in this paper already has sufficient practical and interesting insights to merit consideration for publication.
>
> We do, however highlight the relevance of the fANOVA work in **Section 5.4**.
> Moreover, we extend **Section 6.1** to suggest this direction for future work: namely to condition on the removal of main effects and interactions that are deemed “harmful”. This could be achieved through analysis of our publicly available data, and then reanalysing pipelines.
>
> ## S2: Use Bayesian hyperparameter tuning. Rigorously justify your hyperparameter tuning process by using Bayesian optimization.
> As stated in our response to C6: we do not perform HPO. However, use of Bayesian HPO would be interesting in a more computationally expensive study.
>
> ## S3: Include modern approaches. Specifically, consider using a large pretrained text-embedding model with optional parameter-efficient fine-tuning.
> We discuss inclusion of more featurizers in the general comment. We include AoE-BERT [1] as a recent state-of-the-art adaptation of BERT.

---

> > ### Comment · Reviewer_NUE6 · 2025-07-08
> > **Rebuttal Response**
> >
> > Thank you for addressing some of my concerns. In addition to your direct response to me, I have also reviewed the responses to other reviewers, the general rebuttal, and the changes to the manuscript itself.
> >
> > Of the items I highlighted, I understand that you addressed C2, C3, and C4.
> >
> > **C1:** I am willing to accept the author's explanation for limiting to two datasets, particularly since the scope of the paper has been refocused around price regression exclusively.
> >
> > **C5 and C6:**
> > I am not convinced by the authors' response to these issues. This paper is positioned as a comprehensive and rigorous empirical analysis of NLP pipelines for price regression on textual data, however the limitations of C5 (uniform sampling) and C6 (lack of hyperparameter tuning) fundamentally undermine the extent to which the conclusions from this paper hold.
> >
> > In their rebuttal to C5, the authors posit that uniformly sampling configurations is a strength because it exposes "atypical unstudied combinations" that perform surprisingly well (e.g., BPE + BoW). However, in my original review, I had already hypothesized that these unusual combinations were likely only succeeding due to the **under-tuning of more sensible combinations**, a hypothesis which the authors validated in their response to C6, stating: "**We do not perform any tuning of hyperparameters; using defaults...**" Undertuned baselines is a well-known issue in the literature [[0]](https://dl.acm.org/doi/abs/10.1145/3317287.3328534), and a much more likely explanation as to why these unusual combinations performed well.
> >
> > ***
> >
> > ### Re-evaluation of Contributions
> >
> > This work is positioned as a rigorous empirical analysis of a diverse range of pipeline options for price regression. The authors highlight four main conclusions:
> >
> > * **Observations (i) and (ii):** The findings that contextualized embeddings outperform non-contextual features and that BERT requires fine-tuning are already well-established in the NLP community. These do not represent a novel contribution.
> > * **Observation (iii):** The fact that model performance is sensitive to upstream pipeline choices is a foundational concept in machine learning, and central to the argument. This fact is precisely what makes the distribution over components in the pipeline and rigorous tuning so important. This is not a novel observation, but rather a pitfall that the analysis presented in the paper has fallen victim to.
> > * **Observation (iv):** The conclusion that vector embeddings perform worse than BoW for GBDT models is not rigorously supported. It is a known that complex embedding-based models require careful tuning to outperform simpler, robust baselines like BoW with GBDTs. In fact, this exact observation has been made previously in [[1]](https://proceedings.neurips.cc/paper_files/paper/2022/hash/0378c7692da36807bdec87ab043cdadc-Abstract-Datasets_and_Benchmarks.html). Attributing this performance difference to the components themselves, rather than to the lack of tuning, is a misinterpretation of the results.
> >
> > Based on this analysis, the paper does not accomplish the goal it set out to perform. The work is presented as providing "quantitative, and rigorous, evidence" to challenge assumptions, but the use of uniform sampling and lack of hyperparameter tuning makes the presented evidence too weak to be actionable. The claim of analyzing "out-of-the-box performance" does not salvage the study, as practitioners do not randomly combine components and deploy them with default settings when seeking optimal performance. The entire premise of drawing general conclusions about which components are "best" is invalidated by this experimental design.
> >
> > [0] Lipton, Zachary C., and Jacob Steinhardt. "Troubling Trends in Machine Learning Scholarship: Some ML papers suffer from flaws that could mislead the public and stymie future research." Queue 17.1 (2019): 45-77.
> >
> > [1] Grinsztajn, Léo, Edouard Oyallon, and Gaël Varoquaux. "Why do tree-based models still outperform deep learning on typical tabular data?." Advances in neural information processing systems 35 (2022): 507-520.

---

### Author Response · Authors · 2025-06-16
**General comment**

We would like to thank the reviewers, and AE, for their time. The reviews are detailed and help us to improve the strength of the submission. We were especially happy to see that reviewers (yETR/NbcV) found our work actionable and practically relevant. We also enjoyed reading that NUE6 appreciated our efforts to make the work accessible to a wide body of researchers through a clear and broad background.

There were some comments common to more than one reviewer; and we address those comments here. Comments individual to a specific reviewer are addressed in a reply to said review.

We provide a key to highlight the reviewers and corresponding weaknesses/requests for  change that correspond to each given general point, i.e. 'yETR W2/C2'  indicates reviewer yETR mentioned this in Weakness 2 and Requested  Change 2.

---

> ### Author Response · Authors · 2025-06-16
> **General comment**
>
> # More datasets
>
> yETR W2/C2 | NbcV W1/C1 | NUE6 W1/C1
>
> All reviewers had a comment requesting more datasets. We agree that the study of only two datasets is a limitation of the work, which we have explicitly added to **Section 6.1** of the paper. In this reply we will initially address this point more generally.  Then comment on suggested alternatives by different reviewers, to summarize our thoughts in one place.
>
> Broadly the small number of datasets is a consequence of few compelling publicly available datasets. As noted in the paper, it is relatively easy to envisage industry applications: for example in car insurance where one may predict the damage associated with an accident from a description. However, typically this form of high quality data is commercially sensitive and thus not publicly available. This is also perhaps a consequence of the rapidly changing landscape of ML research and that a relatively new paradigm (textual regression) has not had sufficient time to develop a large array of high quality public datasets.
> We believe that while studying two datasets is admittedly few, it is a reasonable first step and the use of rigorous statistical approaches reduces the potential for misleading conclusions.
> Notably, reviewers also highlighted the fact both datasets are price regression tasks, in light of this we propose to change the title and references to “price regression” instead of generally “regression” to more clearly highlight the focus of the paper. Furthermore, we add more explicit discussion on the limitations of studying only 2 datasets to Section 6.1.
> Having focused on specifically price regression, we have opted to not include an extra price regression dataset; instead prioritising inclusion of an alternative featurizer and model.
>
>
> Reviewer yETR suggested datasets in review rating prediction and student essay scoring. We believe review rating prediction is essentially a multi-class extension of sentiment classification, not regression. We wish this work to focus on problems which are compelling applications of specifically price regression.
> With regards student essay scoring, we did consider this when originally working on our submission. However, we decided that whilst a very interesting application, we believe essay scoring ought be evaluated not only on accuracy but also crucially on explainability - which is outside the scope of this study; and that the two should not be disconnected for ethical reasons [1].
> Preliminary analysis on the essay scoring data (not presented in this work) on the accuracy of an essay scoring dataset were broadly similar - but there was insufficient time to investigate the implications for explainability and we did not cover as many pipelines as studied for our chosen datasets.
> Reviewer NbcV also suggested reframing existing classification tasks as regression, i.e. reviews rating predictions, as with yETR's suggestion we do not believe the recasting of classification to be a compelling application of regression. Likewise, SemEval Task 7 involves prediction of an artificial “intensity” metric, that while having some justification in it's derivation (via a Best-Worst Scaling), lacks a compelling “ground truth” to regress to.
>
> # More featurization algorithms
>
> yETR W1/C1 | NUE6 W5/S3
>
> Two reviewers noted that the most “modern” method studied, BERT, is already considered relatively old in the space of NLP models. We agree that adding more modern featurisers would make this work more compelling and so extend our analysis to include the use of the AoE-BERT embeddings [2] suggested by yETR. We also include the finetuning of this featurizer. However, we note that BERT, although older,  is still widely used (having had >70mil downloads last month on HF); and is the backbone for many of the more modern approaches (as mentioned in MTEB [3]). Therefore, we believe the analysis and conclusions relating to BERT approaches are still relevant and timely.
> Given the timeframe of the rebuttal period, it would not be feasible to try a substantially larger embedding model; but given the excellent performance of AOE-BERT in MTEB we believe it to be a useful extension.
> None of the conclusions of this paper were altered through inclusion of AOE-BERT; and a discussion on the specific comparison of vanilla BERT to AOE-BERT is presented in Appendix C.
>
> # Normalize metrics
>
> NbcV W3/C3 | NUE6 C4
>
> Two reviewers suggested normalizing metrics to enable greater contextualization of the results. Whilst any affine scaling will not change the conclusions, we agree it eases comprehension to not have such disparate scales. We divide the reported metrics by the RMSE obtained by using a training set mean as a constant prediction. This effectively shows the improvement the pipeline achieves over the simplest possible baseline: where no signal is extracted from the text. For clarity: 0.8 on this scale corresponds to a 20% improvement over a constant prediction.

---

> > ### Author Response · Authors · 2025-06-16
> > **References**
> >
> > ## References
> >
> > [1] Aloisi, C., 2023. The future of standardised assessment: Validity and trust in algorithms for assessment and scoring. European Journal of Education, 58(1), pp.98-110.
> > [2] Li, Xianming, and Jing Li. "AoE: Angle-optimized embeddings for semantic textual similarity." Proceedings of the 62nd Annual Meeting of the Association for Computational Linguistics (Volume 1: Long Papers). 2024.
> > [3] Muennighoff, Niklas, et al. "MTEB: Massive Text Embedding  Benchmark." Proceedings of the 17th Conference of the European Chapter  of the Association for Computational Linguistics. 2023

---

### Decision · Action_Editor_weq5 · 2025-08-01

**Recommendation:** Reject

**Additional Comments:**

This ambitious paper attempts to answer some interesting questions but falls short of doing that well. The improvements made by the authors in response to the reviews were appreciated, but they did not address the key issues outlined above.

**Audience:**

No

**Audience Explanation:**

The paper sets out to answer questions of interest to some NLP practitioners, but some unfortunate choices in the experimental protocol result in answers to different, much less relevant, questions.
The top-10 pipeline results reported in the paper are perhaps its most interesting contribution, but the lack of hyperparameter tuning makes them much less interesting than they could have been.

It would be worth reporting the number of parameters for different parametric model / input representations combinations, as these could be helpful for understanding the results better.

**Claims And Evidence:**

No

**Claims Explanation:**

The authors perform an extensive empirical exploration of NLP pipelines by generating numerous pipeline configurations by making a random choice for each stage, and evaluating their performance. They draw conclusions about the effect of the per-stage choices based on the average performance of sampled pipelines containing these choices. As reviewers pointed out, there are two problems with this experimental protocol:
1. There was no hyperpameter tuning, which is especially problematic in this setting as there is little reason to think that the default hyperparameters would have worked well for the many atypical pipelines generated with random sampling.
2. The effect of a particular choice for a pipeline stage was measured by averaging over the randomly generated pipelines compatible with it. Taking maximum instead of mean would make a lot more sense here, as most practitioners are interested in the pipeline choices that lead to the best performance.
The claims were further weakened by using only two relatively small datasets for the experiments.

**Resubmission Of Major Revision:**

The authors may consider submitting a major revision at a later time.